# The Role of Basic Psychological Needs in the Adoption of Healthy Habits by Adolescents

**DOI:** 10.3390/bs13070592

**Published:** 2023-07-14

**Authors:** Adrián Mateo-Orcajada, Raquel Vaquero-Cristóbal, Juan Pablo Rey-López, Raúl Martín-Campoy, Lucía Abenza-Cano

**Affiliations:** 1Facultad de Deporte, UCAM Universidad Católica de Murcia, 30107 Murcia, Spain; amateo5@ucam.edu (A.M.-O.); labenza@ucam.edu (L.A.-C.); 2Faculty of Health Sciences, International University of Valencia (VIU), 46002 Valencia, Spain; 3IES Felipe VI, 30510 Murcia, Spain; raul.martin2@murciaeduca.es

**Keywords:** autonomy, adherence to Mediterranean diet, competence, physical activity level, relatedness

## Abstract

Previous research in this field has not examined the significance of each of the basic psychological needs (BPNs) on changes in the physical activity level, adherence to the Mediterranean diet (AMD), kinanthropometric and derived variables, and the physical fitness of adolescents. Therefore, the purpose of this study was (a) to examine the variances in physical activity, AMD, and kinanthropometric and derived variables, as well as fitness levels, among adolescents with varying degrees of satisfaction regarding each of the BPNs and (b) to assess the differences in the study variables among adolescents based on whether the BPNs are satisfied individually or jointly. The sample consisted of 791 adolescents (404 males and 387 females; average age: 14.39 ± 1.26 years old). The findings indicated that adolescents in the highest percentiles (75–100) of competence, autonomy, or relatedness showed higher scores in physical activity and AMD and better kinanthropometric and physical fitness variables than adolescents in the lowest percentiles (0–25). Adolescents who showed joint satisfaction of all BPNs showed the best results on all variables analyzed. In addition, it should be noted that competence played the most relevant role.

## 1. Introduction

Self-determination theory (SDT) defines autonomy, competence, and relatedness as the three basic psychological needs (BPNs) [1]. Satisfaction of BPNs in adolescents is necessary to achieve well-being and decrease vulnerability to the development of any psychopathology [2]. Adolescents who have all BPNs satisfied together seem to show adequate health status, although individual satisfaction of each BPN has also been shown to influence psychological and behavioral development. [3]. Thus, the results found in previous research showed that the joint satisfaction of the BPNs was related to autonomous motivation, favoring enjoyment and less pressure, with relatedness being the most determinant BPN in this relationship. Competence predicted controlled motivation positively, while autonomy and relatedness predicted it negatively; in addition, competence was negatively related to the feeling of pressure [3].

In addition, it should be noted that in previous cross-sectional research, the authors showed the relationship between LBW and certain healthy lifestyle habits, such as the practice of physical activity [4] or adherence to an optimal nutritional pattern [5]. Thus, with respect to the practice of physical activity, adolescents who had more barriers to physical activity practice showed higher BPN frustration [4]. Similarly, the BPNs were significantly related to adherence to an optimal nutritional pattern, specifically adherence to the Mediterranean diet (AMD), with the AMD score being higher in adolescents who had greater satisfaction of the BPNs, although it is unknown whether the differences were in competence, autonomy, or relatedness, since the authors calculated a final score for the satisfaction of the BPNs without specifying in which of these needs the differences were found [5].

Furthermore, the preliminary findings from pilot studies conducted by the researchers have identified additional personal factors that play a crucial role in maintaining good health and the prevention of chronic diseases in adolescents, such as kinanthropometric and derived variables [6] and the maintenance of a high level of physical fitness [7]. The latter also showed a positive relationship with BPNs, thus becoming a fundamental aspect of the integral health of the adolescent population. Regarding kinanthropometric and derived variables, in previous studies, the researchers demonstrated that autonomy was highest in males when body mass index (BMI) was around 18.50. For females, the study findings revealed that autonomy exhibited an upward trend as body size discrepancies became less negative. Autonomy reached its peak and stabilized when the discrepancy reached +1 [6]. Regarding fitness variables, only one previous study analyzed this relationship, showing that fitness improvements in adolescents could be increased in environments where the satisfaction of BPNs was favored, mainly competence and relatedness [7]. However, it is important to note that these studies had limitations; for example, Markland and Ingledew [6] only included BMI as a variable to assess body composition, while in the research by Erturan-Ilker et al. [7], the measurement of specific physical fitness tests was not provided as a fitness score was calculated, preventing the analysis of the association of the specific performance in these tests to the satisfaction of BPNs. These reasons call for further research to address anthropometric and fitness variables in relation to BPN satisfaction.

In the psychological field, the importance of joint or individual satisfaction with BPNs has been analyzed [3]. However, no previous studies have analyzed whether differences in the level of physical activity, AMD, kinanthropometric and derived variables, and fitness depend on the degree of satisfaction with each of the BPNs. Likewise, it has not been analyzed whether one of the BPNs in particular plays a more important role than the rest in the changes that occurred in these variables [3]. Therefore, the purpose of this study was (a) to examine the variances in physical activity, AMD, kinanthropometric and derived variables, and fitness among adolescents with different degree levels of satisfaction of each of the BPNs and (b) to assess the differences in the level of physical activity, AMD, kinanthropometric and derived variables, and physical fitness of adolescents according to whether the BPNs are satisfied individually or jointly.

Based on the objectives of the present research, as well as previous scientific literature, the following research hypotheses (H) are proposed: (H1) adolescents who experience higher levels of satisfaction with BPNs are expected to exhibit increased levels of physical activity, greater AMD, better physical performance, and better kinanthropometric and derived variables; and (H2) in the absence of previous research that would allow us to establish an accurate hypothesis as to which BPN is more relevant to the study variables, we hypothesize that adolescents who have all BPNs satisfied collectively will demonstrate elevated levels of physical activity, AMD, and physical performance and improved kinanthropometric and derived variables.

## 2. Materials and Methods

### 2.1. Design

The current study employed a cross-sectional design and utilized non-probability convenience sampling. Four compulsory secondary schools in various regions of the Region of Murcia, Spain, were involved in the study. The research adhered to the STROBE statement [8], guiding its design and manuscript development. Furthermore, the study received prior approval from the Institutional Ethics Committee of the Catholic University of Murcia (code: CE022102). The research protocol and design adhered to the guidelines set forth by the World Medical Association and the Helsinki declaration. According to the psychologists’ code of conduct, the ethical principles of beneficence and non-maleficence, fidelity, and responsibility, integrity, justice, and respect for the rights and dignity of people were followed during all processes of the research [9].

### 2.2. Participants

The final sample consisted of 791 adolescents (404 males and 387 females) aged 12–16 years old (mean age: 14.39 ± 1.26 years old). These adolescents willingly participated in the research after providing informed consent forms signed by both themselves and their parents. The sample size was determined using the methodology employed in previous studies [10] and the statistical software Rstudio 3.15.0 (Rstudio Inc., Boston, MA, USA). This calculation was based on the standard deviations (SD) obtained from earlier research conducted on BPNs in the adolescent population [11]. Thus, for an SD = 0.77 for autonomy, SD = 0.82 for competence, and SD = 0.85 for relatedness, and an estimated error (d) of 0.07 for autonomy and 0.08 for competence and relatedness, the minimum sample size required to extrapolate the results with a 99% confidence interval was 764 adolescents.

The participants included in the study satisfied the following inclusion criteria: (a) completion of all questionnaires, kinanthropometric measurements, and physical tests in their entirety; (b) not presenting any disease or surgical operation that prevented participation in any of the tests; (c) age range between twelve and sixteen years old; and (d) attending compulsory secondary education.

### 2.3. Instruments

#### 2.3.1. Measurement of BPNs

To assess competence, autonomy, and relatedness (BPNs), the Spanish version of the Basic Psychological Needs Scale (BPNS) [12] was employed. This scale consists of 18 items, with 6 items dedicated to each dimension. Participants rated each item on a Likert scale ranging from 1 to 6 points (1 totally false; 6 totally true). Consequently, a score between 6 and 36 points was obtained for each dimension.

Based on previous research [13], the analysis of the degree of satisfaction of each of the BPNs was determined by establishing the 0–25, 25–50, 50–75, and 75–100 percentiles. Thus, the score for each dimension was as follows: competence (percentile 0–25: <22; percentile 25–50: <28; percentile 50–75: <32; percentile 75–100: ≥32), autonomy (percentile 0–25: <22; percentile 25–50: <26; percentile 50–75: <30; percentile 75–100: ≥30), and relatedness (percentile 0–25: <21; percentile 25–50: <25; percentile 50–75: <29; percentile 75–100: ≥29).

In addition, BPNs were considered as satisfied when the score was above the 50th percentile, resulting in the following classification: none (no BPN satisfied), competence (competence satisfied only), autonomy (autonomy satisfied only), relatedness (relatedness satisfied only), competence and autonomy satisfied, competence and relatedness satisfied, autonomy and relatedness satisfied, and all BPNs satisfied.

#### 2.3.2. Physical Activity Level and AMD Measurement

To evaluate the participants’ physical activity level, the Spanish version of the “Physical Activity Questionnaire for Adolescents” (PAQ-A) was employed [14,15]. This questionnaire consists of nine items, and the final score is calculated as the average of the scores from the first eight items, resulting in a value between 1 and 5 points [15].

The assessment of the participants’ adherence to the Mediterranean diet was conducted using the “Mediterranean Diet Quality Index for children and adolescents” (KIDMED) questionnaire [16]. This questionnaire comprises 16 items, with 12 items reflecting positive aspects (+1) and 4 items reflecting negative aspects (−1). The scoring range for this questionnaire is from 0 to 12 points.

#### 2.3.3. Kinanthropometric and Derived Variables Measurement

In accordance with the protocol established by the International Society for the Advancement of Kinanthropometry (ISAK) [17], three anthropometrists (levels 2 to 4) conducted measurements using standardized techniques. The measurements included three basic parameters: body mass, height, and sitting height. Additionally, three skinfold measurements were taken (triceps, thigh, and calf), along with five girth measurements (relaxed arm, waist, hips, thigh, and calf).

The instruments utilized for the measurements were as follows: a TANITA BC 418-MA Segmental (TANITA, Tokyo, Japan) (body mass); a SECA stadiometer 213 (SECA, Hamburg, Germany) (height and sitting height); a Harpenden skinfold caliper (Burgess Hill, UK) (skinfolds); and an inextensible tape, Lufkin W606PM (Lufkin, Missouri City, TX, USA) (girths).

The derived variables calculated with the final values of the kinanthropometric measurement were: BMI, waist-to-hip ratio (waist girth/hip girth) [18], corrected arm girth (arm relaxed girth − (π × triceps skinfold)), corrected thigh girth (thigh girth − (π × thigh skinfold)), corrected calf girth (calf girth − (π × calf skinfold)), Σ3 skinfolds (triceps, thigh, and calf), fat mass (%) [19], and muscle mass [20].

#### 2.3.4. Physical Fitness Measurement

In accordance with the methodology employed in prior studies [21], the following physical fitness tests were performed: 20 m shuttle run test to measure cardiorespiratory capacity and prediction of maximal oxygen consumption (VO2 max.) [22,23]; handgrip strength for the measurement of upper limb strength using a Takei Tkk5401 digital handheld dynamometer (Takei Scientific Instruments, Tokyo, Japan) [24]; sit-and-reach to assess hamstring flexibility using an Acuflex Tester III box (Novel Products, Rockton, IL, USA); countermovement jump (CMJ) to assess lower limb explosive strength by means of a force platform with a sampling frequency of 200 Hz (MuscleLab, Stathelle, Norway) with which jump height was measured [25]; and a 20 m sprint to measure running speed using single-beamed photocells (Polifemo Light, Microgate, Italy) placed at hip height [26].

### 2.4. Procedure

Data collection for this research took place in each of the high schools, specifically during the physical education classes. The tests were conducted in a covered sports pavilion to minimize the potential interference of confounding variables. It is important to note that all tests were performed on the same day to maintain consistency in data collection.

The protocol, based on previous research [27], consisted of (1) the completion of the BPNS, PAQ-A, and KIDMED questionnaires; (2) kinanthropometric measurements taken by anthropometrists; (3) a single attempt of the sit-and-reach test conducted before the warm-up to mitigate the potential influence of warm-up activities on test performance; (4) a warm-up session consisting of 5 min of progressive running and 10 min of joint mobility exercises; (5) familiarization and explanation of the handgrip, CMJ, and 20 m sprint tests to adolescents; and (6) the handgrip strength, CMJ, and 20 m sprint tests. The order of execution of these tests was randomized for each adolescent. Each test was repeated twice, with the best value recorded. The rest period between each attempt of the same test was two minutes, while five minutes were provided between different tests, in line with the recommendations of the National Strength and Conditioning Association (NSCA) regarding fatigue management [28]; (7) a single attempt of the 20 m shuttle run test was conducted. Four experienced researchers were assigned to oversee the execution of the tests. Each researcher was responsible for conducting the same test throughout all measurements.

### 2.5. Data Analysis

Initially, the normality of the data was evaluated using the Kolmogorov–Smirnov test, as well as an analysis of skewness and kurtosis. As the data demonstrated a normal distribution, parametric tests were employed for subsequent analyses. Descriptive statistics, including the mean (M) and standard deviation (SD), were utilized to summarize the analyzed variables. The internal consistencies of self-reported measures were evaluated using Cronbach’s alpha. Subsequently, four one-way ANOVA tests were conducted. The first ANOVA aimed to identify differences in the study variables among different percentiles in the competence variable. The second ANOVA focused on the autonomy variable, while the third ANOVA examined the relatedness variable. Lastly, the fourth ANOVA was conducted to determine differences in the study variables based on the BPNs satisfied by the adolescents. For statistically significant variables, Bonferroni’s pairwise comparison was employed. The effect size (ES) was calculated using partial eta squared (η^2^), with values classified as small (ES ≥ 0.10), moderate (ES ≥ 0.30), large (ES ≥ 1.2), or very large (ES ≥ 2.0). The significance level was set at *p* < 0.05 [29]. All tests performed were evaluated for statistical significance at this level. The SPSS statistical package (v.25.0; SPSS Inc., Chicago, IL, USA) was utilized for the statistical analysis.

## 3. Results

Prior to the analysis, the internal consistencies of the self-reported measures were calculated. Thus, the BPN survey reported a Cronbach’s alpha between 0.768 and 0.820 (competence: 0.768; autonomy: 0.820; and relatedness: 0.776), the PAQ-A reported 0.840, and the AMD reported 0.756. A scale has an acceptable internal consistency when the Cronbach’s alpha value is greater than 0.70 [30], which indicates a high internal consistency for the present study.

Table 1 shows the differences in the studied variables according to the percentile of satisfaction of the competence variable. The results show significant differences in the physical activity score, AMD, height, individual-corrected girths, the sum of three skinfolds, fat mass, muscle mass, VO2 max, handgrip strength, CMJ, and the 20 m sprint.

The post hoc analysis of the differences between the satisfaction percentiles of the competence variable is shown in Appendix A. Thus, with respect to the level of physical activity practiced, the lowest percentiles (0–25 and 25–50) showed lower scores as compared to the highest percentiles. For AMD, the 0–25 percentile showed significantly lower scores as compared to the rest of the percentiles. Similarly, in the kinanthropometric and derived variables, the 75–100 percentile showed significant differences as compared to the 0–25 and 25–50 percentiles, with the 75–100 percentile score being higher in height, sitting height, corrected girths, and muscle mass but lower in the sum of three skinfolds and fat mass. Regarding the physical fitness tests, performance was higher in adolescents in the 75–100 percentile as compared to the rest of the percentiles in VO2 max, handgrip strength, CMJ, and the 20 m sprint, except for adolescents in the 50–75 percentile.

The differences in the studied variables, according to the percentile of satisfaction of the autonomy variable, are found in Table 2. The results showed significant differences in physical activity level, AMD, height, corrected girths, muscle mass, VO2 max, handgrip strength, CMJ, and the 20 m sprint.

Appendix A shows the post hoc analysis of the differences between the percentiles of satisfaction of the autonomy variable. The level of physical activity practiced was significantly lower in the 0–25 percentile as compared to the rest of the percentiles, as well as in the 25–50 percentile as compared to the 75–100. Regarding AMD, the score of adolescents in the 75–100 percentile was significantly higher than that of adolescents in the 0–25 percentile. According to the kinanthropometric and derived variables, it is noteworthy to find that adolescents in the 0–25 percentile showed significantly lower scores as compared to the rest of the percentiles in height, corrected girths, and muscle mass. In physical fitness variables, adolescents in the 0–25 percentile showed significantly lower performance in VO2 max, handgrip strength, CMJ, and the 20 m sprint, as compared to the 50–75 and 75–100 percentiles, as well as in CMJ and 20 m sprint, as compared to the 25–50 percentile.

Table 3 shows the differences in the studied variables according to the percentile of satisfaction of the relatedness variable. The results showed significant differences in the level of physical activity practiced, AMD, height, muscle mass, VO2 max, handgrip, CMJ, and the 20 m sprint.

The subsequent post hoc analysis (Appendix A) showed that adolescents in the 75–100 percentile had a higher level of physical activity as compared to the rest of the percentiles, as well as adolescents in the 50–75 percentile as compared to the 0–25 percentile. In AMD, the score was significantly lower in the 0–25 percentile compared to the 50–75 and 75–100 percentiles. Regarding the kinanthropometric and derived variables, adolescents in the 0–25 percentile showed lower scores in height than the rest of the percentiles, as well as in muscle mass as compared to the 50–75 and 75–100 percentiles. According to the physical fitness variables, performance was significantly lower in adolescents in the 0–25 percentile as compared to those in the 50–75 and 75–100 percentiles for VO2 max, handgrip, CMJ, and the 20 m sprint. Furthermore, in VO2 max and CMJ, the differences were also significant between the 25–50 and 75–100 percentiles, with the score of adolescents in the 75–100 percentile being higher in both tests.

The differences in the studied variables according to the BPNs satisfied in the adolescents are shown in Table 4. The results showed significant differences between the groups analyzed in the level of physical activity practiced, AMD, height, corrected arm girth, corrected thigh girth, sum of three skinfolds, fat mass, muscle mass, VO2 max, handgrip strength, CMJ, and the 20 m sprint.

It is worth noting that the level of physical activity was significantly lower in adolescents with no basic psychological needs satisfied, as compared to those with satisfied competence, competence and autonomy, competence, and relatedness, or all BPNs. In addition, adolescents with satisfied competence showed higher scores than those with satisfied relatedness, while those with satisfied competence and relatedness showed higher scores than those with satisfied autonomy and relatedness. On the other hand, adolescents who had all BPNs satisfied showed higher scores than those who only had autonomy, relatedness, or autonomy and relatedness satisfied. Regarding the AMD, adolescents with no BPNs satisfied showed lower scores than those with competence and relatedness satisfied or those with all BPNs satisfied (Appendix A).

The results from the kinanthropometric and derived variables showed that adolescents who did not have any BPNs satisfied had lower scores than adolescents who had competence and autonomy satisfied in height, corrected arm girth, and muscle mass and lower scores than adolescents who had competence and relatedness satisfied in corrected thigh girth and muscle mass but higher scores in the sum of three skinfolds and fat mass, as well as lower scores than adolescents who had all BPNs satisfied in height, corrected thigh girth, and muscle mass (Appendix A).

As for the physical condition variables, performance was significantly lower in VO2 max, handgrip, CMJ, and the 20 m sprint in adolescents who had no BPNs satisfied, as compared to those who had competence together with autonomy or relatedness, as well as compared to those who had all BPNs satisfied. In addition, it is worth noting that in VO2 max, adolescents who had satisfied competence showed higher scores than those who did not have any BPNs satisfied; in addition, adolescents who had satisfied autonomy and relatedness showed lower performance than those who had competence and relatedness or all BPNs satisfied (Appendix A).

## 4. Discussion

The objectives established for the present research were (a) to determine the differences in the level of physical activity, AMD, kinanthropometric and derived variables, and fitness among adolescents with different degree levels of satisfaction of each of the BPNs and (b) to establish the differences in the level of physical activity, AMD, kinanthropometric and derived variables, and physical fitness of adolescents according to whether the BPNs are satisfied individually or jointly. The results indicate that adolescents who practice more physical activity have a greater satisfaction of the BPNs, which is similar to the results of previous research [31]. However, what is particularly noteworthy is that significant differences in the level of physical activity were observed specifically between the highest percentiles of the relatedness variable (50–75 and 75–100). Thus, the group of adolescents in the 75–100 percentile showed the highest score. However, this did not occur in the competence and autonomy variables, as there were no significant differences between the 50–75 and 75–100 percentiles in the level of physical activity. A possible explanation for these results is that during adolescence, peers have a great influence on the behaviors and habits acquired [32], with the presence of friends or parents who practice sports and with whom they can practice being some of the main factors that influence the sports practice of the adolescent population [33]. This is one of the main incentives for the practice and permanence of sports at this age and could also explain why adolescents with higher satisfaction in the relatedness variable have a higher level of physical activity.

In addition, another novelty found in the present investigation is the determinant role played by the satisfaction of the need for competence in the practice of physical activity during adolescence. Thus, adolescents who had all BPNs satisfied jointly or those who had satisfied competence individually or together with autonomy or relatedness practiced more physical activity as compared to those who did not have any satisfied BPNs or satisfied autonomy or relatedness individually. This could be due to the fact that a greater perception of competence would lead adolescents to seek optimal challenges to maintain or improve their skills [34], with sports practice being an ideal setting that, if the demands are adapted to the level of the participant, allows this need to be satisfied [35]. Another possible explanation would be that competence has been shown to be the most relevant BPN in dropping out of sports practice in the adolescent population due to its influence on the sense of sports accomplishment [36]. Therefore, it is logical to think that adolescents who drop out of sports practice have a lower satisfaction of this BPN. These results point to the importance of proposing challenging, stimulating, and individualized activities in the sports environment and in physical education classes, which are also very achievable by adolescents, to favor the satisfaction of the need for competence and, with it, the practice of physical activity.

Regarding AMD, the lowest percentile (0–25) of satisfaction of the competence, autonomy, or relatedness variables showed a significantly lower adherence to these nutritional guidelines compared to the other percentiles. However, no significant differences were found in the level of AMD in any of the BPNs when comparing the other BPN satisfaction percentiles (25–50 with 50–75; 50–75 with 75–100; and 25–50 with 75–100). In previous research, the authors showed a significant and positive relationship between BPN satisfaction and AMD [5], as well as the importance of satisfying BPNs to facilitate the adoption of healthy lifestyle habits [37]. The present research goes further and shows the differences in the level of AMD considering the degree of satisfaction of each of the BPNs, establishing that adolescents with a reduced level of satisfaction of competence, autonomy, or relatedness are at risk of having a low level of AMD, with the negative effects on physical and psychological health that this could entail [38]. This information is crucial and highly relevant to the well-being of the adolescent population. It highlights that a minimum level of satisfaction with BPNs is necessary to promote the adoption of healthy lifestyle habits, including AMD.

Similar to the practice of physical activity, differences were also found in AMD according to individual or joint satisfaction of BPNs. Adolescents who did not have any satisfied BPNs scored lower in AMD as compared to adolescents who had satisfied all BPNs and those who had competence and relatedness jointly satisfied. A possible explanation for these results would be that the acquisition of a certain healthy lifestyle habit favors the acquisition of other healthy habits [39], with the BPNs being of great importance in the adoption of these habits [37], so that adolescents with satisfied competence and who had a high level of physical activity practice are also likely to have high AMD. And, with respect to relatedness, the satisfaction of this BPN in adolescents who had greater AMD could be due to the fact that healthy behaviors in adolescents are influenced by the social network to which they belong, with many of the changes in these behaviors being promoted by changes in social status or the popularity that they entail [40] or also by the fundamental role played by parents in the adoption of eating habits, with adolescents living at home with their parents having the highest AMD [41]. Therefore, the results obtained show that competence and relatedness jointly satisfied seem to exert an important influence on AMD, but this will need to be corroborated in future studies.

According to the kinanthropometric and derived variables, adolescents in the lower percentiles of satisfaction (0–25) of the BPNs showed a lower height and sitting height, greater body fat accumulation, and less muscle development than adolescents in the higher percentiles (50–75 and 75–100). Authors of previous research have shown that autonomy is related to body mass and body size discrepancies [6], but these same authors did not analyze differences in kinanthropometric and derived variables in terms of BPN satisfaction. The results found could be explained by the fact that in the present society, body image is a determining factor in the psychological satisfaction of individuals [42] and is especially relevant during adolescence, given the rise of insecurities related to one’s physique [43] as well as critical judgments by peers that are granted special importance [44]. Furthermore, it should be added that the maturation process is different in each adolescent, with those who mature earlier being taller, with greater muscle mass and higher testosterone production [45], which is especially relevant, as physical dimension is one of the most important in the configuration of the general self-concept [46] and is strongly related to the satisfaction of BPNs [47].

In fact, adolescents who had satisfied all the BPNs, or competence individually or together with autonomy or relatedness, were those who showed better kinanthropometric and derived variables, as compared to adolescents who did not have any satisfied BPNs. This could be due to the fact that a good physical self-concept improves the perception of competence and is related to the measurement of subjective psychological well-being in the adolescent population [11]. Another possible explanation could be that a higher self-concept is also related to increased physical activity in adolescents [48], with less fat mass accumulation in adolescents who are more physically active [49]. However, further research is warranted to elucidate the relationship between BPNs and kinanthropometric and derived variables, as self-concept is greatly modified during adolescence, especially in girls [50].

Regarding physical fitness, adolescents in the lowest percentile (0–25) of BPN satisfaction showed worse performances in VO2 max, handgrip strength, CMJ, and 20 m sprint tests, as compared to the highest satisfaction percentiles (50–75 and 75–100). In addition, competence seemed to be the most important BPN for physical fitness, as adolescents who had all BPNs satisfied, or those with satisfied competence independently or together with autonomy or relatedness, showed better performances. Only one previous research study is known in which the authors analyzed the relationship between BPNs and the physical fitness of adolescents, and although it showed that environments that satisfy BPNs may favor increased fitness, it did not include differences in specific physical fitness tests, as it only calculated a total fitness score [7]. The results of this study are groundbreaking as they reveal significant differences in physical tests such as the 20 m shuttle run test, handgrip strength, CMJ, or 20 m sprint according to the degree of satisfaction of BPNs and that the satisfaction of competence seemed to be the most important for performance in the physical fitness test. A possible explanation for these results would be that better performance in physical fitness tests is related to higher motor competence and performance in certain sports modalities [51]. In previous research, the authors concluded that athletes who performed better were those who scored higher on the BPNs (ranked in the highest percentiles of satisfaction for each BPN) [52] and mainly those with the highest satisfaction of the need for competence because they perceived that they could overcome their challenges to a greater extent [53].

The findings from this investigation provide support for the first research hypothesis (H1), as they indicate that adolescents who experience higher levels of satisfaction with BPNs demonstrate a higher level of physical activity, greater AMD, better physical performance, and better kinanthropometric and derived variables, and it was the adolescents in the highest percentiles of each of the BPNs who showed higher scores in each of the study variables. The results of this study confirm the acceptance of the second research hypothesis (H2). The findings demonstrate that adolescents who have all BPNs collectively satisfied exhibit a higher level of physical activity, AMD, physical performance, and improved kinanthropometric and derived variables. In addition, the results obtained allow us to give greater relevance to the need for competence, since when not all BPNs were satisfied, the satisfaction of competence alone or together with relatedness or autonomy showed significant differences.

This research is similar to those previously carried out in the adolescent population [21,27,54] but has a particularly relevant novelty. The analysis of the satisfaction of BPNs individually or jointly is something that has not previously been carried out in adolescents. This finding holds significant relevance due to the demonstrated importance of BPNs in adolescent development. BPNs play a crucial role in acquiring healthy habits and enhancing body composition and physical fitness. However, prior to this study, it remained unclear whether individual satisfaction with a specific BPN was associated with these improvements or if the collective satisfaction of all BPNs was the key factor. The results highlight the importance of joint satisfaction with all BPNs for the observed enhancements, providing valuable insights into the interplay between BPN satisfaction and various aspects of adolescent well-being. This study provides evidence in this regard and adds to previous studies on the importance of physical activity and adherence to the Mediterranean diet for adolescent development, focusing especially on the psychological field and, more specifically, on the BPNs.

Regarding the limitations of the present study, it should be noted that, unlike other psychological scales, the BPNS (Basic Psychological Needs Scale) does not have a classification scale to determine the degree of satisfaction of each of the BPNs, so percentiles had to be established according to the scores of the adolescents in the sample. The single use of questionnaires is an aspect to be considered in future research, since more information on the satisfaction of the BPNs would be obtained through the combined use of the questionnaire and interviews or focus groups on the adolescent population. Since this was a cross-sectional study, causality cannot be established in the results found, so longitudinal studies are required to establish the indicated relationships between the BPNs and other psychological variables, as well as between BPNs and changes in the level of physical activity, AMD, kinanthropometric and derived variables, and physical fitness.

## 5. Conclusions

To conclude, the present study shows that adolescents in the highest percentiles (50–75 and 75–100) of BPN satisfaction had better levels of physical activity, AMD, kinanthropometric and derived variables, and physical fitness. Furthermore, it is noteworthy that adolescents who had all BPNs satisfied showed higher levels of physical activity, AMD, better kinanthropometric and derived variables, and better physical fitness, with competence being the most relevant; when not all BPNs were satisfied, the satisfaction of competence alone or together with autonomy or relatedness reported the greatest benefits.

The relevance of the present research lies in the fact that it establishes the importance that BPNs can have in the healthy development of the adolescent population beyond the sports field in which this relationship had been widely studied. Thus, the results show the relevance of the satisfaction of the BPNs in the adoption of healthy lifestyle habits, which is fundamental in a society in which the adolescent population has considerably reduced its practice of physical activity and its adherence to adequate nutritional patterns. Therefore, this research may be of relevance to all professionals working with adolescents (coaches, teachers, medical staff, and even parents). 

## Figures and Tables

**Table 1 behavsci-13-00592-t001:** Differences in the physical activity score, AMD, kinanthropometric and derived variables, and physical fitness variables according to the degree of satisfaction of the competence variable.

Variable	Descriptors (M ± SD)				F	*p*	Effect Size (η^2^)
	Competence (0–25) (*n* = 158)	Competence (25–50) (*n* = 225)	Competence (50–75) (*n* = 167)	Competence (75–100) (*n* = 241)			
Physical activity score	2.24 ± 0.66	2.56 ± 0.54	2.78 ± 0.58	2.92 ± 0.65	41.003	<0.001	0.142
AMD	5.73 ± 2.64	6.48 ± 2.16	6.97 ± 2.38	6.89 ± 2.36	9.100	<0.001	0.035
Body mass (kg)	56.27 ± 14.20	56.66 ± 13.57	57.36 ± 12.18	57.81 ± 12.77	0.513	0.673	0.002
Height (cm)	161.78 ± 8.50	162.30 ± 9.21	164.48 ± 8.59	164.75 ± 9.04	5.252	0.001	0.021
BMI (kg/m^2^)	21.44 ± 4.29	21.41 ± 4.14	21.10 ± 3.50	21.20 ± 3.69	0.293	0.830	0.001
Sitting height (cm)	84.45 ± 4.32	84.39 ± 6.33	85.81 ± 4.45	85.71 ± 4.63	4.276	0.005	0.017
Waist girth (cm)	69.23 ± 9.30	69.88 ± 9.19	69.48 ± 7.90	69.96 ± 8.67	0.271	0.847	0.001
Hip girth (cm)	91.04 ± 9.67	90.68 ± 9.41	90.56 ± 8.41	90.73 ± 9.07	0.073	0.974	0.001
Waist-to-hip ratio	0.76 ± 0.05	0.77 ± 0.05	0.77 ± 0.05	0.77 ± 0.06	1.433	0.232	0.006
Corrected arm girth (cm)	20.76 ± 2.97	21.01 ± 2.93	21.57 ± 2.94	21.80 ± 3.06	4.868	0.002	0.019
Corrected thigh girth (cm)	39.03 ± 4.49	39.58 ± 4.96	40.47 ± 4.70	41.31 ± 5.12	8.124	<0.001	0.032
Corrected calf girth (cm)	28.79 ± 2.67	29.14 ± 3.90	29.52 ± 2.98	29.81 ± 3.02	3.496	0.015	0.014
Sum of 3 skinfolds (cm)	57.38 ± 27.27	54.07 ± 26.27	49.56 ± 23.76	47.15 ± 22.42	6.173	<0.001	0.024
Fat mass (%)	25.15 ± 11.37	23.85 ± 10.70	21.82 ± 9.67	21.00 ± 9.14	6.276	<0.001	0.025
Muscle mass (kg)	17.60 ± 4.64	18.38 ± 4.95	19.33 ± 5.10	20.11 ± 5.30	8.773	<0.001	0.034
VO2 max. (mL/kg/min)	36.87 ± 4.91	38.77 ± 5.06	40.71 ± 6.08	41.62 ± 5.67	26.596	<0.001	0.097
Handgrip right arm (kg)	24.15 ± 6.83	25.41 ± 7.63	28.16 ± 8.32	28.72 ± 9.07	13.125	<0.001	0.050
Handgrip left arm (kg)	22.30 ± 6.10	23.65 ± 6.84	26.20 ± 7.88	26.75 ± 8.16	14.614	<0.001	0.056
Sit-and-reach (cm)	15.87 ± 8.93	15.26 ± 8.57	16.48 ± 8.53	16.05 ± 8.98	0.622	0.601	0.003
CMJ (cm)	20.92 ± 5.71	22.62 ± 6.61	24.58 ± 7.17	25.79 ± 7.19	18.399	<0.001	0.069
20 m sprint (s)	4.16 ± 0.39	4.00 ± 0.58	3.83 ± 0.61	3.78 ± 0.45	19.469	<0.001	0.073

AMD: adherence to Mediterranean diet; BMI: body mass index; VO2 max: maximum oxygen consumption; CMJ: countermovement jump.

**Table 2 behavsci-13-00592-t002:** Differences in the physical activity score, AMD, kinanthropometric and derived variables, and physical fitness variables according to the degree of satisfaction of the autonomy variable.

Variable	Descriptors (M ± SD)				F	*p*	Effect Size (η^2^)
	Autonomy (0–25) (*n* = 189)	Autonomy (25–50) (*n* = 167)	Autonomy (50–75) (*n* = 208)	Autonomy (75–100) (*n* = 227)			
Physical activity score	2.38 ± 0.65	2.63 ± 0.63	2.69 ± 0.64	2.85 ± 0.62	18.200	<0.001	0.068
AMD	6.05 ± 2.51	6.40 ± 2.46	6.64 ± 2.25	7.03 ± 2.33	5.866	0.001	0.023
Body mass (kg)	55.21 ± 15.07	56.85 ± 11.11	58.60 ± 13.71	57.41 ± 12.19	2.140	0.094	0.009
Height (cm)	161.05 ± 9.06	163.70 ± 8.72	164.14 ± 9.12	164.48 ± 8.64	5.746	0.001	0.023
BMI (kg/m^2^)	21.17 ± 4.46	21.19 ± 3.58	21.63 ± 3.99	21.15 ± 3.54	0.689	0.559	0.003
Sitting height (cm)	84.23 ± 4.51	85.35 ± 4.85	85.46 ± 4.69	85.33 ± 6.03	2.312	0.075	0.009
Waist girth (cm)	68.58 ± 9.59	69.54 ± 7.90	70.78 ± 9.07	69.73 ± 8.36	1.993	0.114	0.008
Hip girth (cm)	89.76 ± 10.32	90.25 ± 7.88	91.74 ± 9.29	90.99 ± 8.76	1.667	0.173	0.007
Waist-to-hip ratio	0.76 ± 0.05	0.77 ± 0.05	0.77 ± 0.05	0.77 ± 0.06	0.651	0.582	0.003
Corrected arm girth (cm)	20.59 ± 2.91	21.45 ± 2.99	21.66 ± 2.98	21.51 ± 3.04	4.759	0.003	0.019
Corrected thigh girth (cm)	38.94 ± 4.86	40.39 ± 5.23	40.59 ± 4.88	40.71 ± 4.68	5.181	0.002	0.020
Corrected calf girth (cm)	28.63 ± 3.57	29.56 ± 3.51	29.67 ± 2.93	29.52 ± 2.96	4.084	0.007	0.016
Sum of 3 skinfolds (cm)	55.30 ± 27.51	49.80 ± 24.87	52.39 ± 25.98	49.30 ± 21.92	2.238	0.083	0.009
Fat mass (%)	24.19 ± 11.47	22.45 ± 10.55	22.93 ± 10.31	21.81 ± 8.86	1.822	0.142	0.007
Muscle mass (kg)	17.54 ± 4.92	19.22 ± 4.91	19.43 ± 5.22	19.51 ± 5.13	6.207	<0.001	0.024
VO2 max. (mL/kg/min)	38.30 ± 5.66	39.27 ± 5.21	40.02 ± 5.78	40.80 ± 5.85	6.812	<0.001	0.027
Handgrip right arm (kg)	24.40 ± 7.36	26.48 ± 7.60	27.63 ± 8.56	28.10 ± 8.88	7.634	<0.001	0.030
Handgrip left arm (kg)	22.55 ± 6.96	24.60 ± 6.99	25.88 ± 7.48	26.06 ± 8.09	8.886	<0.001	0.035
Sit-and-reach (cm)	15.38 ± 8.45	15.36 ± 9.26	16.14 ± 7.92	16.43 ± 9.35	0.719	0.541	0.003
CMJ (cm)	21.40 ± 6.59	23.91 ± 7.51	23.68 ± 6.63	25.36 ± 6.76	10.970	<0.001	0.042
20 m sprint (s)	4.12 ± 0.42	3.95 ± 0.44	3.86 ± 0.64	3.81 ± 0.53	12.995	<0.001	0.050

AMD: adherence to Mediterranean diet; BMI: body mass index; VO2 max: maximum oxygen consumption; CMJ: countermovement jump.

**Table 3 behavsci-13-00592-t003:** Differences in the physical activity score, AMD, kinanthropometric and derived variables, and physical fitness variables according to the degree of satisfaction of the relatedness variable.

Variable	Descriptors (M ± SD)				F	*p*	Effect Size (η^2^)
	Relatedness (0–25) (*n* = 178)	Relatedness (25–50) (*n* = 178)	Relatedness (50–75) (*n* = 217)	Relatedness (75–100) (*n* = 218)			
Physical activity score	2.44 ± 0.71	2.60 ± 0.58	2.63 ± 0.62	2.89 ± 0.63	15.993	<0.001	0.061
AMD	5.80 ± 2.54	6.41 ± 2.34	6.98 ± 2.30	6.89 ± 2.30	9.480	<0.001	0.037
Body mass (kg)	55.71 ± 12.64	58.03 ± 14.67	57.13 ± 12.57	57.38 ± 12.87	0.930	0.426	0.004
Height (cm)	161.22 ± 8.06	163.87 ± 9.68	164.18 ± 9.05	164.03 ± 8.77	4.373	0.005	0.017
BMI (kg/m^2^)	21.44 ± 4.30	21.43 ± 4.04	21.08 ± 3.67	21.25 ± 3.69	0.362	0.780	0.001
Sitting height (cm)	84.32 ± 4.24	85.12 ± 6.74	85.57 ± 4.63	85.27 ± 4.67	1.965	0.118	0.008
Waist girth (cm)	69.26 ± 8.99	70.30 ± 9.50	69.53 ± 8.04	69.72 ± 8.76	0.424	0.736	0.002
Hip girth (cm)	90.61 ± 8.99	90.98 ± 9.72	90.50 ± 9.02	90.89 ± 8.95	0.113	0.952	0.001
Waist-to-hip ratio	0.76 ± 0.06	0.77 ± 0.05	0.77 ± 0.05	0.77 ± 0.06	0.573	0.633	0.002
Corrected arm girth (cm)	20.89 ± 2.84	21.49 ± 3.21	21.45 ± 2.99	21.40 ± 2.97	1.472	0.221	0.006
Corrected thigh girth (cm)	39.04 ± 4.39	40.15 ± 4.91	40.85 ± 5.41	40.49 ± 4.74	4.569	0.054	0.018
Corrected calf girth (cm)	28.99 ± 4.19	29.25 ± 2.94	29.57 ± 2.83	29.53 ± 2.97	1.259	0.287	0.005
Sum of 3 skinfolds (cm)	55.78 ± 27.33	52.62 ± 25.29	48.14 ± 23.74	51.03 ± 23.96	3.002	0.060	0.012
Fat mass (%)	24.35 ± 11.46	23.14 ± 10.50	21.54 ± 9.77	22.54 ± 9.45	2.421	0.065	0.010
Muscle mass (kg)	17.80 ± 4.50	19.14 ± 5.34	19.47 ± 5.22	19.23 ± 5.18	3.879	0.009	0.015
VO2 max. (mL/kg/min)	38.43 ± 6.09	39.01 ± 5.55	40.32 ± 5.74	40.58 ± 5.31	6.102	<0.001	0.024
Handgrip right arm (kg)	24.97 ± 7.46	26.78 ± 8.14	27.62 ± 8.78	27.32 ± 8.42	3.661	0.012	0.015
Handgrip left arm (kg)	23.18 ± 6.71	24.95 ± 7.63	25.72 ± 8.13	25.32 ± 7.40	3.957	0.008	0.016
Sit-and-reach (cm)	15.75 ± 8.56	15.70 ± 8.51	15.98 ± 8.98	16.02 ± 8.94	0.062	0.980	0.001
CMJ (cm)	22.08 ± 6.86	23.01 ± 6.74	24.12 ± 7.22	25.03 ± 6.77	6.418	<0.001	0.025
20 m sprint (s)	4.08 ± 0.55	3.96 ± 0.42	3.87 ± 0.65	3.83 ± 0.44	8.141	<0.001	0.032

AMD: adherence to Mediterranean diet; BMI: body mass index; VO2 max: maximum oxygen consumption; CMJ: countermovement jump.

**Table 4 behavsci-13-00592-t004:** Differences in the physical activity score, AMD, kinanthropometric and derived variables, and physical fitness variables according to the psychological needs satisfied by the adolescents.

Variable	Descriptors (M ± SD)								*p*	Effect Size (η^2^)
	Basic Psychological Needs Met									
	None	Competence	Autonomy	Relatedness	Competence and Autonomy	Competence and Relatedness	Autonomy and Relatedness	All of Them		
Physical activity score	2.38 ± 0.65	2.81 ± 0.61	2.58 ± 0.53	2.38 ± 0.59	2.68 ± 0.68	2.88 ± 0.54	2.51 ± 0.57	2.92 ± 0.61	<0.001	0.126
AMD	5.85 ± 2.55	6.53 ± 2.67	6.27 ± 2.52	6.30 ± 2.22	6.41 ± 2.03	7.14 ± 2.32	6.92 ± 1.82	7.10 ± 2.41	<0.001	0.044
Body mass (kg)	56.13 ± 14.22	55.69 ± 8.61	57.98 ± 14.44	55.00 ± 13.22	58.42 ± 13.95	57.09 ± 13.43	58.18 ± 12.76	57.78 ± 12.36	0.635	0.007
Height (cm)	161.46 ± 8.71	162.60 ± 8.60	162.14 ± 9.01	162.46 ± 9.42	165.41 ± 9.39	164.67 ± 9.33	163.44 ± 8.87	164.70 ± 8.62	0.006	0.027
BMI (kg/m^2^)	21.46 ± 4.50	21.11 ± 3.08	21.85 ± 4.13	20.79 ± 3.56	21.25 ± 3.83	20.82 ± 3.78	21.72 ± 4.06	21.22 ± 3.59	0.750	0.006
Sitting height (cm)	84.55 ± 4.57	84.78 ± 4.33	83.26 ± 9.38	84.64 ± 4.86	86.13 ± 4.67	85.59 ± 5.15	84.76 ± 4.73	85.81 ± 4.41	0.054	0.023
Waist girth (cm)	69.40 ± 9.66	69.44 ± 6.32	70.74 ± 9.82	68.11 ± 8.51	70.17 ± 9.08	68.79 ± 7.84	71.12 ± 8.04	69.91 ± 8.53	0.560	0.008
Hip girth (cm)	90.59 ± 9.81	89.54 ± 7.36	91.79 ± 9.32	89.58 ± 8.69	91.18 ± 9.16	88.84 ± 9.31	92.23 ± 9.70	91.07 ± 8.75	0.455	0.009
Waist-to-hip ratio	0.77 ± 0.05	0.78 ± 0.05	0.77 ± 0.06	0.76 ± 0.05	0.77 ± 0.05	0.76 ± 0.05	0.77 ± 0.06	0.77 ± 0.06	0.760	0.006
Corrected arm girth (cm)	20.76 ± 2.94	21.36 ± 2.33	21.23 ± 3.01	20.85 ± 2.92	22.11 ± 3.42	21.76 ± 3.45	21.12 ± 2.98	21.61 ± 2.87	0.020	0.022
Corrected thigh girth (cm)	38.84 ± 4.68	40.68 ± 3.60	39.93 ± 4.79	39.45 ± 4.76	40.68 ± 4.82	41.86 ± 6.81	40.27 ± 4.98	40.91 ± 4.70	<0.001	0.036
Corrected calf girth (cm)	28.87 ± 4.06	28.79 ± 3.04	29.13 ± 2.86	28.95 ± 2.80	29.87 ± 3.17	30.06 ± 2.94	29.30 ± 2.69	29.69 ± 2.95	0.072	0.017
Sum of 3 skinfolds (cm)	56.73 ± 27.40	51.11 ± 27.82	56.47 ± 27.24	51.10 ± 25.09	47.95 ± 21.23	42.45 ± 20.86	56.00 ± 26.14	49.01 ± 23.12	0.002	0.029
Fat mass (%)	24.86 ± 11.60	22.94 ± 11.98	24.74 ± 10.93	22.92 ± 10.21	20.74 ± 8.36	19.22 ± 8.68	24.47 ± 10.17	21.75 ± 9.33	0.002	0.030
Muscle mass (kg)	17.69 ± 4.69	19.01 ± 3.50	18.44 ± 5.10	18.02 ± 5.04	20.11 ± 5.74	20.50 ± 6.05	18.89 ± 4.76	19.65 ± 5.08	<0.001	0.035
VO2 max. (mL/kg/min)	37.44 ± 5.35	40.81 ± 5.87	38.79 ± 4.97	38.83 ± 4.75	40.78 ± 6.64	41.81 ± 5.00	37.92 ± 4.62	41.34 ± 5.77	<0.001	0.088
Handgrip right arm (kg)	24.30 ± 6.86	26.51 ± 8.09	26.02 ± 7.93	25.27 ± 6.99	29.26 ± 8.87	28.44 ± 9.23	25.20 ± 8.50	28.56 ± 8.74	<0.001	0.053
Handgrip left arm (kg)	22.56 ± 6.41	25.19 ± 8.12	24.14 ± 6.55	23.21 ± 6.91	27.10 ± 8.15	26.08 ± 7.86	23.65 ± 6.65	26.64 ± 8.06	<0.001	0.057
Sit-and-reach (cm)	15.67 ± 8.54	17.31 ± 8.59	16.45 ± 8.71	13.75 ± 9.06	14.60 ± 8.31	15.38 ± 9.45	16.40 ± 8.69	16.76 ± 8.80	0.196	0.013
CMJ (cm)	20.91 ± 6.22	24.32 ± 9.23	22.41 ± 6.06	23.27 ± 6.34	25.74 ± 6.10	26.08 ± 8.01	22.80 ± 6.38	25.13 ± 7.01	<0.001	0.071
20 m sprint (s)	4.14 ± 0.40	3.91 ± 0.55	3.98 ± 0.74	4.03 ± 0.39	3.82 ± 0.37	3.80 ± 0.46	3.95 ± 0.66	3.78 ± 0.57	<0.001	0.075

AMD: adherence to Mediterranean diet; BMI: body mass index; VO2 max: maximum oxygen consumption; CMJ: countermovement jump.

## Data Availability

The data sets generated during and/or analyzed during the current study are available from the corresponding author upon reasonable request.

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
