# Peer review of "The Role of Basic Psychological Needs in the Adoption of Healthy Habits by Adolescents"

_behavsci, 2023, doi:10.3390/bs13070592_

Round 1

Reviewer 1 Report

Thank you for this paper. The exploration of our basic psychological needs is important in maintaining our health through exercise and nutrition habits. Addressing this in adolescence is especially important to develop life long participation in exercise. 

Mainly the revisions needed are with wording clarity. I have attached you document with highlighted phrases and sentences. Please remember that you should personify a noun - a study can't show - research can't find... Researchers can... authors can...  

Line 66 - run on sentence

Your purpose statements were confusing and somewhat unclear. These need to be more specific and written concisely. Remove jargon. 

LINE 289 - these statements were confusing. Be specific as to what you are you summarizing from the results. 

Please review grammar and wording of the highlighted text.  

Thank you for this paper. The exploration of our basic psychological needs is important in maintaining our health through exercise and nutrition habits. Addressing this in adolescence is especially important to develop life long participation in exercise. 

Mainly the revisions needed are with wording clarity. I have attached you document with highlighted phrases and sentences. Please remember that you should personify a noun - a study can't show - research can't find... Researchers can... authors can...  

Line 66 - run on sentence

Your purpose statements were confusing and somewhat unclear. These need to be more specific and written concisely. Remove jargon. 

LINE 289 - these statements were confusing. Be specific as to what you are you summarizing from the results. 

Please review grammar and wording of the highlighted text.  

Author Response

Reviewer 1

Thank you for this paper. The exploration of our basic psychological needs is important in maintaining our health through exercise and nutrition habits. Addressing this in adolescence is especially important to develop life long participation in exercise. 

+ Thank you very much for taking the time to review the manuscript. We will take all your suggestions to improve the manuscript.

Mainly the revisions needed are with wording clarity. I have attached you document with highlighted phrases and sentences. Please remember that you should personify a noun - a study can't show - research can't find... Researchers can... authors can...  

+ Thank you very much for your great contribution. Sentences stating "research showed" or similar have been rewritten.

Line 66 - run on sentence

+ Thank you for your input. This sentence has been split to make it easier to read and understand.

Your purpose statements were confusing and somewhat unclear. These need to be more specific and written concisely. Remove jargon. 

+ Thank you very much for your input. We have rewritten the aims and they have been changed in the abstract and discussion.

LINE 289 - these statements were confusing. Be specific as to what you are you summarizing from the results. 

+ Thank you for your comment. This paragraph has been rewritten for better understanding.

Please review grammar and wording of the highlighted text.

+ Thank you very much for providing the PDF that has facilitated the review.

+ Dear reviewer, again, thank you for taking the time to review the manuscript. We have addressed all your suggestions and believe that you have substantially improved the quality of the manuscript.

Reviewer 2 Report

The aim of the study was a) to determine the differences in the level of physical activity, AMD, kinanthropometric and derived variables, and physical fitness, according to the degree of satisfaction of competence, autonomy, or relatedness; and b) to establish which BPN is more determinant for the study variables, according to the differences in these variables as a function of the joint or individual BPNs satisfaction. I would like to congratulate and thank them for their effort and motivation involved in this research study. The presentation of the research is well documented, with a scientific basis and respects the latest standards regarding the highest level scientific publications. The methodology was chosen correctly. The conclusions support and result from the research and open new directions for future research. The submitted work is interesting and essentially exhausts the subject under discussion. I have only a few minor suggestions:

What the research hypotheses for the study were? Did the study confirm them? Were these results expected? This should be further elaborated at the end of the introduction.

Furthermore, the article states that the study was conducted in accordance with the Declaration of Helsinki and World Medical Association, what is extremely important. However, there is no information whether the participants were treated ethically according to the unified American Psychological Association code of ethics? Please complete this information in the manuscript. This is an important element from an ethical point of view.

Supplementing the article with the above-mentioned scope will in my opinion make a real chance for publication in Behavioral Sciences. I keep my fingers crossed for the final success of the publication. 

Author Response

Reviewer 2

The aim of the study was a) to determine the differences in the level of physical activity, AMD, kinanthropometric and derived variables, and physical fitness, according to the degree of satisfaction of competence, autonomy, or relatedness; and b) to establish which BPN is more determinant for the study variables, according to the differences in these variables as a function of the joint or individual BPNs satisfaction. I would like to congratulate and thank them for their effort and motivation involved in this research study. The presentation of the research is well documented, with a scientific basis and respects the latest standards regarding the highest level scientific publications. The methodology was chosen correctly. The conclusions support and result from the research and open new directions for future research. The submitted work is interesting and essentially exhausts the subject under discussion.

+ Dear reviewer, thank you very much for reviewing our article and making this assessment. We will make the corrections you indicated to improve the quality of the article.

I have only a few minor suggestions:

What the research hypotheses for the study were? Did the study confirm them? Were these results expected? This should be further elaborated at the end of the introduction.

+ Thank you very much for your contribution. Hypotheses have been included at the end of the introduction and contrasted in the discussion.

Furthermore, the article states that the study was conducted in accordance with the Declaration of Helsinki and World Medical Association, what is extremely important. However, there is no information whether the participants were treated ethically according to the unified American Psychological Association code of ethics? Please complete this information in the manuscript. This is an important element from an ethical point of view.

+ Thank you very much for your contribution. We have included this information which is of relevance to the field of psychology.

Supplementing the article with the above-mentioned scope will in my opinion make a real chance for publication in Behavioral Sciences. I keep my fingers crossed for the final success of the publication. 

+ Thank you very much for your contribution. We have included after the conclusions a paragraph mentioning how relevant research can be in the different areas.

+ Thank you very much for contributing to the substantial improvement of the manuscript. We have responded to all your requests and believe that the quality of the manuscript has been enhanced. We remain at your disposal for any other necessary modifications.